# Effect of Supercritical Bending on the Mechanical & Tribological Properties of Inconel 625 Welded Using the Cold Metal Transfer Method on a 16Mo3 Steel Pipe

**DOI:** 10.3390/ma16145014

**Published:** 2023-07-15

**Authors:** Michal Krbata, Robert Ciger, Marcel Kohutiar, Maria Sozańska, Maroš Eckert, Igor Barenyi, Marta Kianicova, Milan Jus, Naďa Beronská, Bogusław Mendala, Martin Slaný

**Affiliations:** 1Faculty of Special Technology, Alexander Dubcek University of Trenčín, 911 06 Trenčín, Slovakia; michal.krbata@tnuni.sk (M.K.); robert.ciger@tnuni.sk (R.C.); maros.eckert@tnuni.sk (M.E.); igor.barenyi@tnuni.sk (I.B.); marta.kianicova@tnuni.sk (M.K.); milan.jus@tnuni.sk (M.J.); 2Faculty of Materials Engineering, Silesian University of Technology, Krasińskiego 8, 40-019 Katowice, Poland; maria.sozanska@polsl.pl (M.S.); boguslaw.mendala@polsl.pl (B.M.); 3Institute of Materials and Machine Mechanics, SAS, Dúbravská Cesta 9/6319, 845 13 Bratislava, Slovakia; nada.beronska@savba.sk; 4Faculty of Mechanical Engineering, Brno University of Technology Technická 2896/2, Královo Pole, 616 69 Brno, Czech Republic; slany.m@fme.vutbr.cz

**Keywords:** tribology, welding, material, superalloy, steel, wear, temperature, hardness

## Abstract

The presented work deals with the investigation of mechanical tribological properties on Inconel 625 superalloy, which is welded on a 16Mo3 steel pipe. The wall thickness of the basic steel pipe was 7 mm, while the average thickness of the welded layer was 3.5 mm. The coating was made by the cold metal transfer (CMT) method. A supercritical bending of 180° was performed on the material welded in this way while cold. The mechanical properties evaluated were hardness, wear resistance, coefficient of friction (COF) and change in surface roughness for both materials. The UMT Tribolab laboratory equipment was used to measure COF and wear resistance by the Ball-on-flat method, which used a G40 steel pressure ball. The entire process took place at an elevated temperature of 500 °C. The measured results show that the materials after bending are reinforced by plastic deformation, which leads to an increase in hardness and also resistance to wear. Superalloy Inconel 625 shows approximately seven times higher rate of wear compared to steel 16Mo3 due to the creation of local oxidation areas that support the formation of abrasive wear and do not create a solid lubricant, as in the case of steel 16Mo3. Strain hardening leads to a reduction of possible wear on Inconel 625 superalloy as well as on 16Mo3 steel. In the case of the friction process, the places of supercritical bending of the structure showed the greatest resistance to wear compared to the non-deformed structure.

## 1. Introduction

Analyses of machine components failures and structures show that their most frequent causes are tribological processes that take place on functional surfaces. The mutual interaction of surfaces is accompanied by the creation of passive resistance to movement, called friction. The result of friction is wear, which is a degradation process that leads to the loss of material from the contact surfaces [1,2]. Friction is associated with the loss of mechanical energy and wear with the loss of material [3]. We can alleviate this situation by better structural arrangement of nodes, change of materials and creation of new surface layers. When creating surface layers for tribological use, we most often encounter welding technology. This enables the application of various welding technologies to create welds from very small thicknesses—in the order of tenths of a millimetre to several tens of centimetres. Currently, nickel-based reactive alloys are widely used as coatings. Welding technology allows not only the restoration of the used surface geometry but also gives them new, oftentimes better properties than the original material [4,5].

In the current analyses of abrasion resistance, the issue of the structure and substructure of welds and their influence on abrasion resistance is not analysed in detail. However, structural and substructural properties can directly affect material removal. Its intensity is conditioned by the strength and cohesive properties of submicroscopic particles just when the surface is stressed by abrasion. For these reasons, it is necessary to design the type of weld based on tribological analysis and surface stress conditions. However, this tribological analysis is impossible for many practical cases due to the imperfection of the developed procedures, which is why even now we use operational and laboratory tests to evaluate the properties of the coatings. Basic scientific knowledge of the coating wear tribology is developed in [3,5,6] and other works. It follows that one of the crucial factors for the intensity of wear in abrasive and erosive wear of coatings is their hardness. The hardness of welds is a function of the chemical composition of the weld and its thermal regime during and after welding. These factors directly affect the structure of the welds. 

Inconel 625 superalloy is suitable for applications that utilize a combination of high strength, corrosion resistance and suitable weldability. The main element used in this superalloy is Nickel. INCONEL^®^ uses the protected designation of a superalloy that is produced under the manufacturer Special Metals Corporation (New Hartford, NY, USA) [7], some other companies engaged in production or sales use their own designations such as: Haynes^®^ 625 [8], VDM^®^ Alloy 625 Nicrofer 6020 hMo [9], ATI 625TM [10], Alloy 625 [11], Altemp 625 [12], IN625 [13] or Inconel 625-0402 [14]. In the presented article, we will use the designation Inconel 625. This designation is widely used, which is evidenced by a wide range of articles devoted to the study of this superalloy [15,16,17,18,19,20,21,22,23]. The nominal range of individual elements in the chemical composition for marking INCONEL^®^ 625 must meet the following (wt.%): Ni (rest, 58.00 min), Cr (20.00–23.00), Mo (8.00–10.00), Nb plus Ta (3.15–4.15), Fe (≤5.00), Ti (≤0.40), Al (<0.40), Co (<1.0), C (<0.10), Si, Mn (each < 0.50), S and P (each < 0.015) [7]. The mechanical strength of the subject superalloy, which can be either in cast or ductile form, is primarily dependent on the solid solution of the γ phase matrix, in which the main alloying elements are Cr, Mo and Nb. The Inconel 625 alloy can be defined as a superalloy that is strengthened by a solid solution and its additional strengthening can be caused by precipitation, the formation of carbide or intermetallic phases. Inconel 625 superalloy is also widely used for creep and thermal fatigue resistance as well as corrosion resistance caused by hot gases. These properties can be used up to a temperature of 800 °C when the superalloy is subjected to solution annealing. 

16Mo3 steel is commonly used in applications that operate at elevated temperatures up to 600–650 °C. Superalloy Inconel 625, on the other hand, has a significantly higher temperature resistance and can withstand temperatures up to 980 °C. Inconel 625 is known for its outstanding corrosion resistance properties. It exhibits excellent resistance to a wide range of corrosive environments, including acids, seawater and high-temperature gases. 16Mo3 steel also offers good corrosion resistance, but it may not be as resistant as Inconel 625 in severe corrosive conditions. 16Mo3 steel is a low alloy steel with good mechanical properties, including strength and toughness. It provides satisfactory performance in applications where moderate strength is required. Inconel 625, on the other hand, is a high-strength superalloy known for its exceptional mechanical properties at both room and high temperatures. While 16Mo3 steel offers good heat resistance and corrosion resistance for moderate-temperature applications, Inconel 625 surpasses it in terms of temperature resistance, corrosion resistance and mechanical properties, especially at high temperatures [24,25].

These leading properties are also widely used in the chemical and energy industries [8,9]. The production of geometrically complex parts is highly demanding due to increased hardness, poor machinability and also low thermal conductivity [26,27]. The technological process for creating the deposit is carried out using the cold metal transfer (CMT) technology, which results in straight steel pipes with the deposit Inconel 625 (IN625) in the required thickness. These pipes are then bent to achieve the appropriate shape. During the bending process, compression and tension stresses occur. The highest stress values are obtained at supercritical curves to which the given semi-finished products are usually exposed. The objective of this work is to determine how the formation of these stresses affects the kind and development of wear of both experimental materials. In the inner radius of the curve, compressive stresses prevail, which lead to plastic deformation of the material due to the influence of which its strengthening occurs. Conversely, tension stress occurs at the outside radius of the pipe curve, which also causes plastic deformation, but to a lesser extent than in the previous case. Both types of stress cause a change in friction coefficient, wear resistance or local change in hardness and roughness.

The presented article is focused on investigating the influence of supercritical bending on the mechanical–tribological properties of Inconel 625 superalloy welded by the CMT method on a 16Mo3 steel pipe. Mechanical damage due to friction can occur during the practical use of these deposits in combustion sieves of industrial furnaces. The originality of this study lies in the simulation of the friction process at an elevated temperature of 500 °C. The most critical points in the construction of these welded pipes can be the supercritical bending points, where a significant state of stress occurs. It was these locations that were the subject of investigation, which were compared with locations without supercritical bending.

## 2. Materials and Methods

For research and collection of experimental samples, a semi-finished product was used—the bimetallic pipe of 16Mo3 steel with a coating of IN625 alloy with a total diameter of d = Ø 39 mm. The supplier of the semi-finished product is the German company UHLIG Rohrbogen GmbH (Langelsheim, Germany). The wall thickness of the basic steel pipe is 7 mm, while the average thickness of the welded layer is 3.5 mm [6]. The deposition of IN625 material on a steel pipe made of 16Mo3 material was carried out by the Cold Metal Transfer (CMT) method. Subsequently, a supercritical bend of 180° with extended ends was created on the pipe with the coating, the basic dimensions of which are shown in Figure 1 [28,29]. A nickel refractory alloy wire with a diameter of 1 mm was used for welding. IN625 is heat-resistant, creep-resistant nickel alloy (NiCr22Mo9Nb) alloyed with Cr, Mo Nb, Fe and other elements. The alloy was strengthened by substitution, i.e., by dissolving alloying elements in a solid Ni solution [30]; however, due to the presence of Nb and other elements, precipitation strengthening may also be present. 16Mo3 steel is a medium alloy heat-resistant steel for use at higher temperatures and pressures. It is well malleable both hot and cold, resistant to corrosion in a water vapor environment up to 530 °C. In terms of use, it is suitable for energy and chemical equipment. It is supplied in a state after annealing, where the resulting structure is ferrite and pearlite. The basic indicative chemical composition for IN625 and 16Mo3 steel are shown in Table 1.

Figure 1b shows a detail of the location of the areas of performed laboratory measurements of hardness and tribological wear tests. The sample was separated from the pipe blank using abrasive cutting. Subsequently, the sample was cast in thermoplastic to carry out the grinding and polishing process. Grinding was carried out with the help of sandpapers with a roughness of 240, 500, 800 and 1200 μm. Subsequently, the surface was polished with a diamond grinding paste with a grain size of 9, 6 and 3 μm with a pressure of 25 N for 10 min for each abrasive at a speed of 150 rpm. The goal was to achieve a smooth surface ready for hardness measurement and tribological tests. Figure 1c shows the interface between Inconel 625 and Steel 16Mo3.

The basic homogeneous austenitic structure (γ) of the alloy is confirmed by the ternary diagram in Figure 2 [31], where the position of the alloy according to the mass fraction of the main alloying elements is also marked. In addition to grains of the basic γ phase, precipitates of the γ′ and γ″ phases also occur in the structure. In the temperature range of 650 to 875 °C, secondary precipitates of type A_3_B such as, e.g., Ni_3_ (Ti, Al) or Ni_3_ (Nb, Al and Ti), whose presence in the structure after thickening have a negative effect on heat resistance and creep resistance. At higher temperatures, the mentioned precipitates completely dissolve back into the matrix [32,33]. The mentioned effect has the effect of improving the creep properties and extending the life of the material. In the initial stages of creep, substitution strengthening of the Ni solid solution contributes to creep resistance. In other phases (above 875 °C and under the simultaneous action of the load), the precipitation of the M_23_C_6_ carbides type also has a significant effect [34,35,36]. In the as-built Inconel 625 Amato et al. [37] observed precipitates rich in Nb, located along the melt pool boundaries, which were identified as the γ″ phase. The provided results of transmission electron microscopy imaging, microanalysis by energy-dispersive X-ray spectroscopy (EDS) as well as X-ray diffraction (XRD) are not convincing, and it cannot be excluded that the observed precipitates are the δ phase. In turn, Zhang et al. [38], using EDS elemental maps acquired in a scanning electron microscope (SEM), revealed significant segregation of Ni and Cr to dendrite cores as well as enrichment of the interdendritic regions in Nb and Mo in Inconel 625.

Further studies performed by TEM allowed them to confirm the existence of δ and γ″ phases. After annealing at a lower temperature of 800 °C for 1 h, Nb- and Mo-rich precipitates were also found [39]. TEM analysis does confirm the co-existence of γ″ along with δ within the interdendritic regions, as demonstrated by the selected area diffraction patterns of both δ and γ″. A survey of the TEM foil revealed γ″ in multiple regions, suggesting its presence is not an aberration. The γ″-phase was also found in additional TEM foils and other annealing conditions [39]. The microstructure of Inconel 625 achieved by L-DED was investigated by Dinda et al. [40]. The studies were devoted mostly to the characterization of the dendritic microstructure in as-built condition and after annealing at the temperature range 700–1200 °C for 1 h. It was determined that the dendritic microstructure remains stable up to the temperature of 1000 °C. 

To summarise, both the δ and γ″ phases precipitate along locally solute-rich interdendritic regions of the as-built solidification microstructure of IN625 during heat treatment at 870 °C. The formation of γ″ at 870 °C is highly unexpected in the wrought alloy, where γ″ typically forms at temperatures of <750 °C. The enhanced precipitation of γ″ at temperatures considerably higher than expected is due to favourable thermodynamic conditions in the solute-rich interdendritic regions of the solidification microstructure, making the kinetics of γ″-precipitation competitive with those of δ.

Figure 3a is the microstructure of the Inconel 625 weld, observed after bending the welded layer in the places with the maximum compressive stresses. The weld has a homogeneous structure formed by grains of solid solution γ (see ternary diagram in Figure 2). The grains have a dendritic shape and are deformed by the influence of stresses during bending. Microstructural analysis was carried out by the metallographic method using a Keyence light optical microscope (Keyence VHX-7000, Keyence International, Mechelen, Belgium) and image analysis software. The material was processed by a standard process of the metallographic samples’ preparation (grinding, polishing and etching). Since Inconel 625 is a nickel alloy, highly resistant to oxidation and acids, “Glyceregia” etchant with the composition of 50% HCl, 33% glycerol and 17% HNO_3_ was used for etching. The etching time was 40 min. The mentioned etchant and especially the etching time are not suitable for 16Mo3 steel. When these etching conditions were used, therefore, the steel was over-etched, and the areas formed by the steel in the finish were blackened. For this reason, it is not possible to optimally etch the base material and the coating at the same time. The microstructure of the 16Mn3 steel pipe on which the Inconel 625 material was welded was determined after metallographic analysis as ferritic–pearlitic, which is typical for this type of steel [41].

The collection of samples for experimental measurements was carried out as follows: A cut was made through the geometric centre of the pipe, and then samples were taken at three different angles of the pipe bend. Sampling is shown in Figure 1. Samples 1 and 4 were cut at an angle of 15°, samples 2 and 5 at an angle of 90° and samples 3 and 6 at an angle of 165°. Subsequently, all the cut samples were prepared as metallographic samples. With the research method used, it is important that the surface has the lowest possible roughness, preferably below Ra = 0.5, which was achieved by successive grinding and polishing of the sample surfaces. The bimetallic pipe was bent super-critically, which caused uneven stress distribution. The reason was that the bending was not conducted using a bending die, but the pipe was bent around a mandrel with a diameter of 25 mm. Tribological measurements were performed on the UMT TriboLab device (Bruker Austria GmbH, Wien, Austria). The measurement parameters were set to a load of 5 N, and a measurement time of 1200 s. The tribological measurement was set as a linear test and was carried out by the “Ball-on-flat” method, in which a hardened steel ball with a diameter of 4.76 mm with a hardness of 700 HV3 and a material designation of G40 was used as the counterpart of the tribological pair. The given ball moved along the Inconel weld reciprocating at a speed of 10 m/s along a path of 10 mm. A new ball was used for each measurement, which was degreased and dried. Wear evaluation was performed using a confocal microscope (LEXT OLS 5100, Olympus Europa SE and Co. KG, Hamburg, Germany). For a thorough analysis, tribological measurements with the same parameters were also carried out on a basic unbent steel tube of material 16Mo3, which also had a coating of Inconel 625 superalloy.

## 3. Results

### 3.1. Hardness Analysis

First, the hardness results were analysed, which are shown in Figure 4. The average hardness of the experimental material Inconel 625, which was measured on the non-bent part, reached the value of 267 HV1. The deformed Inconel 625 material on the outer radius reached the highest hardness on sample no. 2 and had a value of 459 HV1 (Figure 4a). The increase in hardness was caused by plastic deformation, which caused an average increase in hardness on the outer radius of Inconel 625 by 56% compared to the basic undeformed material. On samples from the inner radius of the tube (Figure 4b), the highest hardness was achieved by sample no. 4 and that is 483 HV1. The average increase in hardness from all three internal samples increased by 63%. When comparing the increase in hardness with respect to the radius of the samples, we observe that the inner radius achieved an increase in hardness compared to the outer radius. As a result, a higher degree of plastic deformation occurred at the inner radius of the bend, which led to an increase in hardness. Thus, the nature of the “tension-outer radius/pressure-inner radius” deformation has an impact on the resulting hardness [42,43,44].

As with the Inconel 625 materials, we also evaluated the hardness of the 16Mo3 steel. The average hardness of the basic un-bent material was 183 HV1. The steel samples that were taken from the outer radius showed almost identical hardness values, reaching an increase in the average hardness after deformation by 36% (Figure 4a). The highest hardness of steel from the samples on the outer radius was measured on sample no. 1 and reached the value of 257 HV1. On the inner radius, the increase in hardness reached the same value of 36% (Figure 4b). The results show that the degree of deformation in 16Mo3 steel is the same both on the outer and inner radius.

### 3.2. Coefficient of Friction Aanalysis (COF)

Figure 5a shows the variation of the friction coefficient at the outer radius of the tube. The average COF value was 0.61 for the given radii of the deformed samples on the nickel alloy. Here, we can observe that the difference between the highest and the lowest COF value was only 0.03. So, we can conclude that the different hardness of the individual samples on the outer radius does not play any role in the COF change. The COF value on the deformed Inconel 625 sample was 0.59. Thus, plastic deformation increases the COF value only very minimally. When observing the second series of samples from the inner radius (Figure 5b), we note that the average COF value was 0.61 for the nickel alloy Inconel 625. The same average COF value was recorded on these samples from the outer radius. Also, as in the previous series of samples, the change in hardness does not change the COF.

The COF values of the 16Mo3 steel at the outer radius showed significantly different results compared to the Inconel 625 superalloy. In Figure 5a, we observe that the COF values of the 16Mo3 steel increased almost linearly with respect to the order of the samples. The average value of COF on non-deformed 16Mo3 steel was 0.53. The highest COF value of 0.57 was achieved on sample no. 3, which achieved the lowest hardness of the outer radius samples. The inner radius showed almost identical COF results on 16Mo3 steel (Figure 5b). Also, the same direction of COF increase was recorded here with respect to the marking of the samples, respectively, the COF value increased with decreasing hardness. 

Overall, it can be concluded that supercritical bending does not cause a change in COF on the Inconel 625 superalloy, and the influence of plastic deformation does not play a role in any change in COF compared to the non-deformed sample.

A comparison of the COF curves of two basic undeformed materials (Inconel 625 + 16Mo3) is shown in Figure 6. The development of the COF curve was divided into two zones. A sharp increase in the COF curve was detected in the first zone A, where the adhesion of both contact surfaces increased. At the same time, there was also an increase in the contact area and destruction of the primary surface layer and the beginning of the formation of an oxidation layer. This first zone spike was observed on both types of materials. The second zone B can be characterized as the stabilization of COF values. In the case of 16Mo3 steel, we can see that there is a gradual slight increase in COF, which is caused by the gradual increase of the contact area depending on the changes in the asperity of the surface [45,46]. The COF in the second zone B of the undeformed Inconel 625 material remained stable because there was a uniform exfoliation of the surface particles in the tribological process and thus a dynamic equilibrium was achieved [47,48]. Average COF values for individual samples were taken from a steady state. Overall, we can observe slightly higher COF values for Inconel 625 material compared to 16Mo3 steel. This result is generally caused by an increase in temperature. Respectively, the temperature during the measurements was constant, but a uniform oxidation layer was formed on the surface of the 16Mo3 steel, which acted as a protective film, over which the friction ball was able to move with less resistance. These results are also proven by numerous literatures that were devoted to linear reciprocating motion [49,50,51,52,53]. Other authors mention the type of oxide Fe_3_O_4_, which adheres better to the surface and acts as a solid lubricant [54,55,56,57,58,59]. On the other hand, on the Inconel 625 super alloy, only larger oxidation islands were formed, which were located only in certain parts of the surface of the friction groove and which, after breaking them, supported the formation of hard abrasive particles. These particles then caused an increase in frictional resistance and thus increased the COF itself.

### 3.3. Wear Rate Analysis

In Figure 7a, the wear of the samples from the outer radius of the pipe was evaluated. Inconel 625 in a mechanically unstressed state shows a higher rate of wear than plastically deformed material. The same reduction in the wear rate on the outer radius also occurred on the 16Mo3 steel due to the use of deformation due to supercritical bending. A comparison of the wear of Inconel 625 and 16Mo3 steel shows a seven times higher wear rate against the superalloy.

As a result of the rotation of the pipe blank through a supercritical bend, an inhomogeneous pressure distribution occurs in the outer radius, where tensile pressures prevail, accompanied by the formation of microcracks, which are the main inhibitors of wear. The compressive forces acting in the inner radius (Figure 7b) cause a greater degree of plastic deformation, where the atomic lattice is compacted without the formation of cracks, which is shown last but not least by more uniform wear of the examined samples.

#### Wear Mechanisms

An overall comparison of the formed grooves’ shape is shown in Figure 8. Here, we can observe that in the case of Inconel 625 superalloy, the boundaries of the grooves were irregular in shape and showed changes in narrowed and widened surfaces. While the 16Mo3 steel showed a uniform groove shape along its entire length, we can also observe a significant change in the total width of the grooves of both materials.

Selected samples, which are shown in Figure 9, were subjected to an in-depth analysis of the wear mechanism. This wear surface analysis was evaluated using a LOM microscope. To compare the wear mechanism, the basic samples of both types of materials, which did not go through the process of deformation due to supercritical bending, were evaluated first. On the base of the undeformed Inconel 625 material (Figure 9a), we observe large grooves that are caused by abrasive particles that arise from microparticles of the released material [60,61]. These microparticles plastically strengthen due to friction and create deep grooves [62,63,64]. Also, on the given sample of the base material, there are integral large oxidation surfaces that show a degree of strengthening, which is proven in the given figure. There are also small pits on the sample surface, which are created by the destruction of released microparticles that are pressed and rolled on the surface of the material and cause local strengthening of the material due to plastic deformation. These microparticles arise from abrasive wear, when the surfaces of both contact materials touch, and micro-joints are created during the friction process, which can be released immediately or later and thus create an abrasive particle. This release takes place above the spot of the microweld, where the material exhibits lower strength [65]. The wear mechanism on the basic undeformed Steel 16Mo3 (Figure 9b) consisted of adhesive wear with the appearance of microcracks in the perpendicular direction of the friction movement of the pressure ball. Slight oxidative wear occurred almost on the entire surface, which can be observed as a gray discoloration of the friction surface. Sample no. 1 of the Inconel 625 material (Figure 9c) showed similar wear mechanisms as the basic non-deformed Inconel 625 material, with the difference that there was an increase in the proportion of oxidative wear due to the increase in hard oxidation surfaces. The number of small pits from abrasive particles has decreased due to the increase in hardness of the material after deformation. There were also craters on the given sample, which were created after the thin, hard oxidation shell was torn off. These thin shells of material can be gradually crushed and transformed into hard abrasive microparticles [66,67]. Figure 9d represents the worn surface of 16Mo3 steel on sample no. 1. On the surface, there were solid oxidation surfaces as well as deep grooves caused by abrasive particles. The occurrence of microcracks was also observed here, as in the basic non-deformed steel. Analysis of the surface of sample no. 4 of the material Inconel 625 shows a significant occurrence of oxidative wear, which is shown by the formation of oxidation islands (Figure 9e). There are significantly plastically deformed areas on these islands. Significant grooves are observed only on the surface that does not show oxidative wear. In the case of Inconel 625 materials, we can, therefore, conclude that increasing the hardness leads to an increase in the rate of oxidative wear. The last Figure 9f shows the worn surface of 16Mo3 steel on sample no. 4. The surface had an evenly distributed oxidation surface as well as deep grooves caused by abrasive particles. The appearance of microcracks was also observed here, as in the basic non-deformed steel.

Oxygen content was evaluated on basic samples that were not deformed using EDS analysis. By comparison, we observed that the oxygen content was 31% on the friction grooves of the Inconel 625 superalloy (Figure 10). This content, as already mentioned, mainly created larger integral islands, which showed a higher degree of strengthening due to the preservation of their elevated position compared to the surrounding surface. In contrast, the basic Steel 16Mo3 (Figure 11) showed an almost uniform occurrence of oxygen on the surface of the friction groove. The total share of O_2_ content was up to 92%. This result relates to the chemical composition of individual materials. Since 16Mo3 steel contains more than 98% Fe, thus supporting the formation of a uniform oxidation layer [68,69,70]. In contrast, Superalloy Inconel 625 contains only a few percent of chemical elements that can promote corrosion, so an even layer of oxidative wear does not form on the surface. The occurrence of oxidation islands is mainly caused by the transfer of Fe from the contact of the pressure steel friction ball material G40. The highest level of oxidation was observed in the formed grooves.

### 3.4. Analysis of Surface Roughness

A comparison of all roughness values of the scraped friction surfaces is shown in Figure 12. The basic texture of the Sa surface, which arose after preparing the samples for tribological tests, had a value of Sa = 0.45 µm. The basic roughness of the undeformed Inconel 625 superalloy after the friction process was 3.27 µm (Figure 12a). After the deformations on the outer radius of the pipe, the average roughness was 2.44 µm. The same reduction in Sa roughness after deformation also occurred on 16Mo3 steels, where the roughness value of the undeformed material was 2.55 µm and decreased to an average value of 1.13 µm. These values on both types of materials from the outer radius of the pipe showed the same surface degradation values for the friction process. Similar results were obtained on the second set of samples from the inner radius of the tube (Figure 12b). Here, Inconel 625 showed a roughness value of Sa = 2.52 µm, except for the last sample no. 6 where the roughness value was the highest of all deformed samples of Inconel 625 superalloy and reached the value of Sa = 3.13 µm. This significant increase is closely related to the hardness, which was the lowest of all Inconel 625 supergas (Figure 4b). The decrease in hardness was caused by a small plastic deformation at the end of the bending process, and thus the material was less resistant to the friction body, which caused more significant wear, which was related to an increase in roughness.

Figure 13 compares the 3D topographic regions of the selected samples. The basic undeformed Inconel 625 superalloy shows a significant change in surface asperity, which consists of many abrasive grooves (Figure 13a). Similar results of a large change in surface texture in the form of high height differences were also recorded on non-deformed 16Mo3 steel. In this sample, perpendicular microcracks caused by adhesive wear were observed at the bottom of the deep crater. After supercritical bending, sample no. 1, which originates from the outer radius of the pipe, softened the overall texture of the surface, which is confirmed by both experimental materials (Figure 13c,d). Inconel 625 shows grooves on its surface, which are caused by abrasive particles (Figure 13c). On sample no. 4 of Inconel 625 material, which is shown in Figure 13e, a significant reduction in the number of abrasive grooves and at the same time an increase in the number of oxidation islands is visible. The smoothing of the surface on the 16Mo3 materials (Figure 13f) was also caused by the strengthening of the material after bending, on which we can observe a combination of abrasive grooves and smooth places, which show signs of moderate oxidative wear, which acts as a solid lubricant and prevents significant degradation of the material.

## 4. Discussion

The use of superalloy Inconel 625 as a material to increase the service life of heat exchangers is known worldwide, as pointed out by the authors [70,71]. The purpose of studying these superalloy deposits is to reduce the cost of their creation and extend their service life. Pipes that are coated using the CMT method are bent with a critical bend, which results in a high degree of deformation. The first hardness measurement results were performed on the semi-finished bent pipe, which clearly demonstrated that due to the high degree of deformation and deformation hardening occurs. The principle of strain hardening is the displacement of dislocations within slip planes and the formation of new dislocations, which prevent further plastic deformation of the material. Strain hardening results in a material with greater levels of internal stresses that affect mechanical properties such as hardness, strength, notch toughness and abrasion resistance [72,73]. Bending causes tensile and compressive stresses. Compressive stresses occur on the inside of the bend, while tensile stresses occur on the outside. The results of the hardness measurement point to the fact that the distribution of these stresses is not uniform within the bend. From the point of view of the increase in hardening, we conclude that the rate of strain hardening for Inconel 625 material is greater than that of 16Mo3 steel. However, for all measured samples, there was an increase in hardness compared to the base undeformed material. Increased hardness has a definite effect on the tribological properties of the measured materials, as proven by other authors [74,75]. The first indicator was the coefficient of friction (COF) results, which showed higher values compared to the base material. In general, higher COF values show more wear than lower COF values [76,77]. This assumption was confirmed when evaluating the results of tribological tests using the amount of material removed. In the entire range of tested samples, there was a reduction in the amount of material removed. Increasing hardness is in practice an effective method of increasing abrasive resistance. After performing the tribological tests, macroscopic photos of the friction grooves were taken. From the evaluation of these photos, it is clear that there is a different wear mechanism for Inconel 625 and 16Mo3 steel. The most significant is the difference in the width of the friction grooves. In order to understand the mechanism of wear, microscopic images of the surface and their topography were made for the purpose of evaluating the roughness of the surface. The results point to two mechanisms of wear formation. In the case of Inconel 625, oxidation islands are formed, which are characterized by high hardness, but when the friction element passes repeatedly, they are damaged and oxidation particles are released, which have the ability to further disturb the surface. When forming friction grooves, 16Mo3 steel creates a continuous oxidation layer in the worn area, which effectively prevents further wear. To confirm these claims, an EDS analysis of the surface was performed, which directly shows the distribution of oxygen on the worn surface of the friction grooves. Due to the effect of deformation strengthening, the roughness of the friction grooves also decreased. Lower roughness results in lower COF values, which can be observed in the measured results.

## 5. Conclusions

The presented article focused on investigating the influence of supercritical bending on the mechanical–tribological properties of the Inconel 625 superalloy welded by the CMT method on a 16Mo3 steel pipe. A steel ball G40 was used as a counterpart. The samples were divided into two groups (outer and inner radius). We can summarize the following conclusions and results from the presented article:The hardness of both types of materials (Inconel 625 and Steel 16Mo3) was increased after the supercritical bending process due to plastic deformation.The highest hardness value of the Inconel superalloy was achieved on sample no. 4, on which the greatest degree of plastic deformation occurs during the bending process. This highest degree of deformation is caused by bending, which is caused by a compressive force on the inner radius of the pipe, and at the same time, the material is supported by a mandrel during bending, which causes additional strengthening in the direction perpendicular to the axis of the pipe.The lowest hardness values were achieved for both radii on samples no. 3 and 6 where the degree of deformation was the smallest because the pipe is almost only aligned at this angle.Deformation due to bending leads to only a minimal increase in COF for both materials. The COF of Inconel 625 reaches higher values compared to 16Mo3 steel. This result is caused by the formation of a more uniform oxidation layer on the surface of the friction groove of 16Mo3 steel, which acts as a solid lubricating film.The location of the supercritical bend shows a reduction in the rate of wear with respect to the basic undeformed material Inconel 625 and also to steel 16Mo3.Superalloy Inconel 625 shows approximately seven times higher wear rate than 16Mo3 steel. Lower COF values for 16Mo3 steel also led to reduced degradation of the friction surface.The friction process causes oxidation wear on Inconel 625 superalloy, which is characterized by the formation of limited oxidation islands. While the 16Mo3 steel shows a uniform distribution of the oxidation layer over the surface.Deformation hardening leads to an increase in the rate of oxidative wear in Inconel 625 friction rods.Strain hardening after the friction process reduces the roughness of the friction groove, compared to the basic non-deformed experimental materials Inconel 625 and 16Mo3.Superalloy Inconel 625 shows predominantly abrasive wear, which is supported by the formation of hard microparticles that are released from coarse oxidation islands. A combination of abrasive, adhesive and oxidative wear occurs on 16Mo3 steel, which with increasing strength leads to a predominance of adhesive and oxidative wear with a small occurrence of abrasive grooves.

Overall, it can be concluded that strain hardening leads to a reduction of possible wear on Inconel 625 superalloy as well as on 16Mo3 steel. This strain hardening is largest at the beginning of the supercritical bend location in the inner bend radius. In the event of a possible occurrence of the friction process, the rate of wear on the supercritical bends of individual parts of the structure will show the greatest resistance.

## Figures and Tables

**Figure 1 materials-16-05014-f001:**
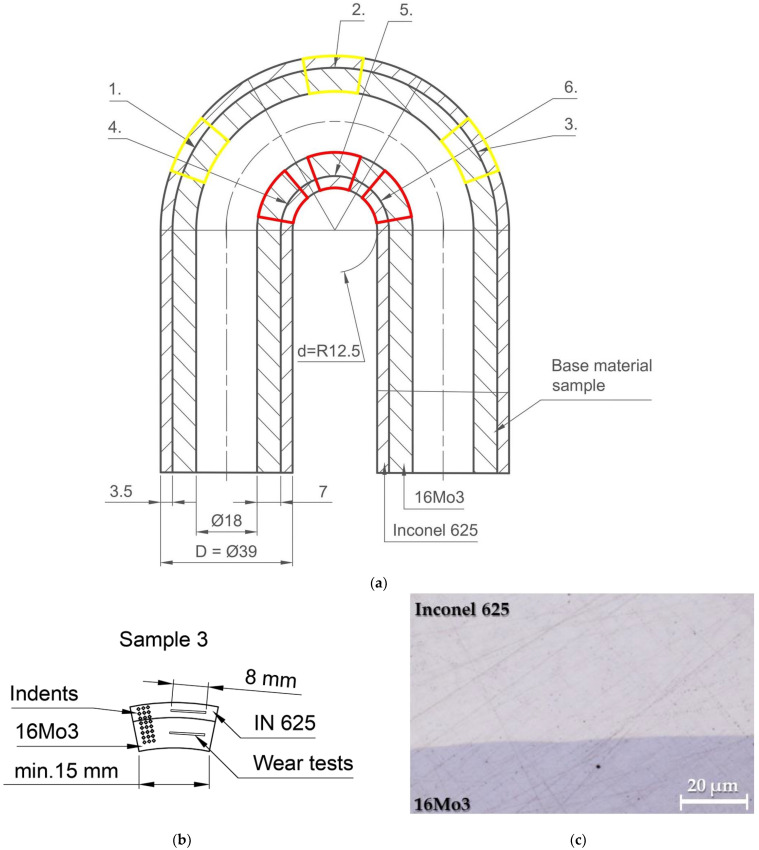
(**a**) The pipe bend used for the experiment with the marked collection points of individual samples (no. 1–6); (**b**) detail the location of the areas of performed laboratory measurements; (**c**) interface between Inconel 625 and Steel 16Mo3.

**Figure 2 materials-16-05014-f002:**
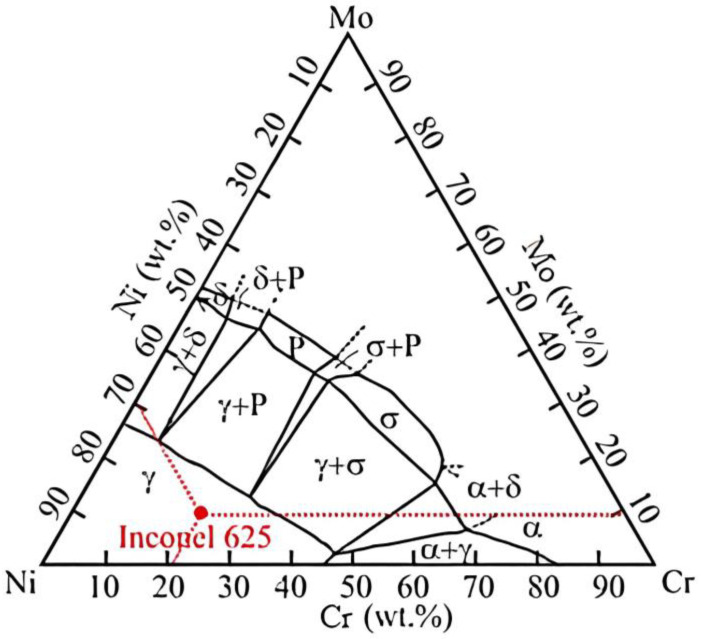
NiCrMo ternary diagram and Inconel 625 superalloy position at the room temperature; *α*—phase of solid solutions; γ—basic homogeneous austenitic structure; δ—orthorhombic intermetallic phase; *P*—phase belongs to the group of topologically close-packed (TCP) phases [31].

**Figure 3 materials-16-05014-f003:**
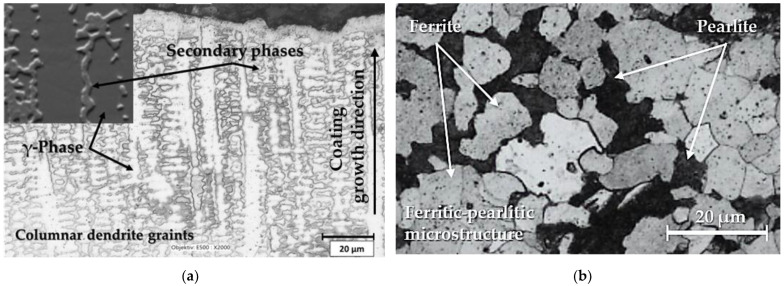
Microstructure from base material: (**a**) Inconel 625; (**b**) 16Mo3 steel.

**Figure 4 materials-16-05014-f004:**
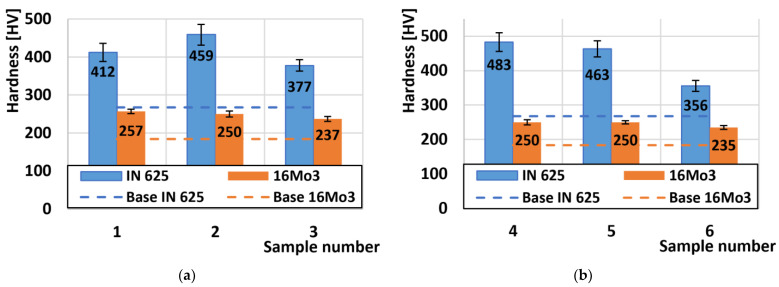
Comparison of hardness results: (**a**) samples (no. 1–3) on the outer radius of the pipe; (**b**) samples (no. 4–6) on the inner radius of the pipe.

**Figure 5 materials-16-05014-f005:**
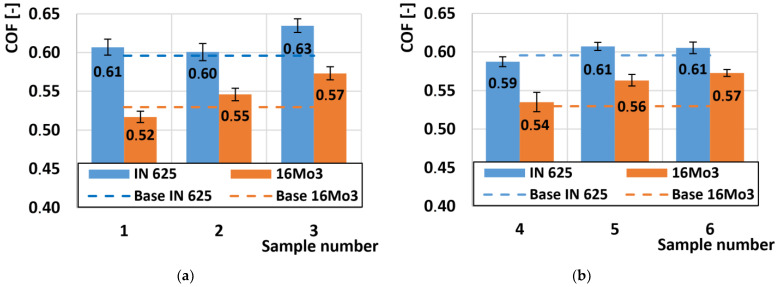
Comparison of COF: (**a**) samples (no. 1–3) at the outer radius of the pipe; (**b**) samples (no. 4–6) on the inner radius of the pipe.

**Figure 6 materials-16-05014-f006:**
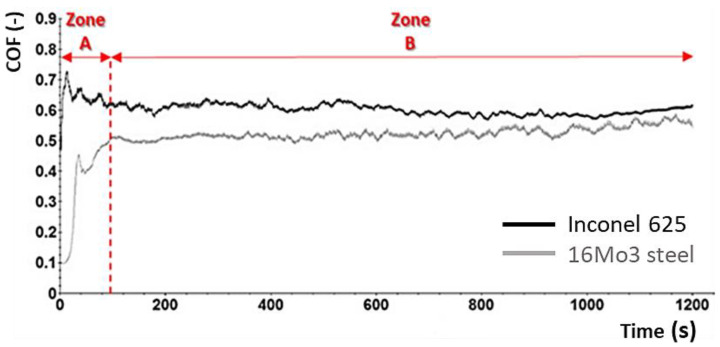
Comparison of COF curves of undeformed samples of Inconel 625 and 16Mo3 steel; Zone A—a sharp increase in COF due to an increase in the adhesion of the contact surfaces; Zone B—stabilization of COF values due to uniform contact of friction surfaces.

**Figure 7 materials-16-05014-f007:**
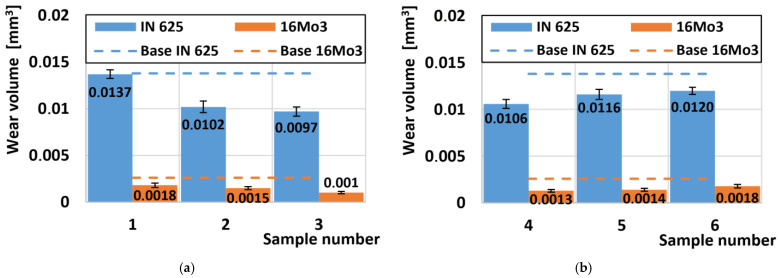
Comparison of wear: (**a**) samples (no. 1–3) at the outer radius of the pipe; (**b**) samples (no. 4–6) on the inner radius of the pipe.

**Figure 8 materials-16-05014-f008:**
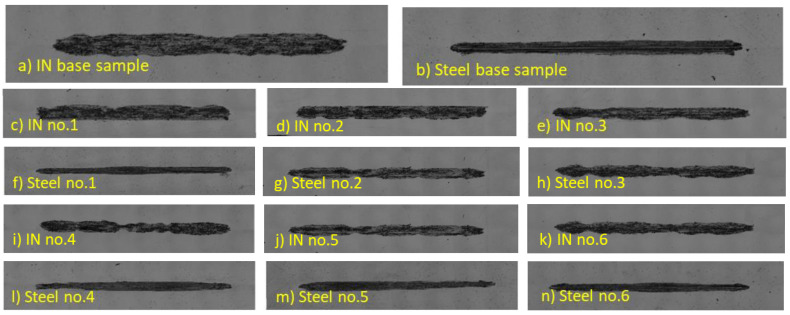
Comparison of friction groove shapes: (**a**) Inconel 625 base sample; (**b**) 16Mo3 steel base sample; (**c**) Inconel 625 no. 1; (**d**) Inconel 625 no. 2; (**e**) Inconel 625 no. 3; (**f**) 16Mo3 steel no. 1; (**g**) 16Mo3 steel no. 2; (**h**) 16Mo3 steel no. 3; (**i**) Inconel 625 no. 4; (**j**) Inconel 625 no. 5; (**k**) Inconel 625 no. 6; (**l**) 16Mo3 steel no. 4; (**m**) 16Mo3 steel no. 5; (**n**) 16Mo3 steel no. 6.

**Figure 9 materials-16-05014-f009:**
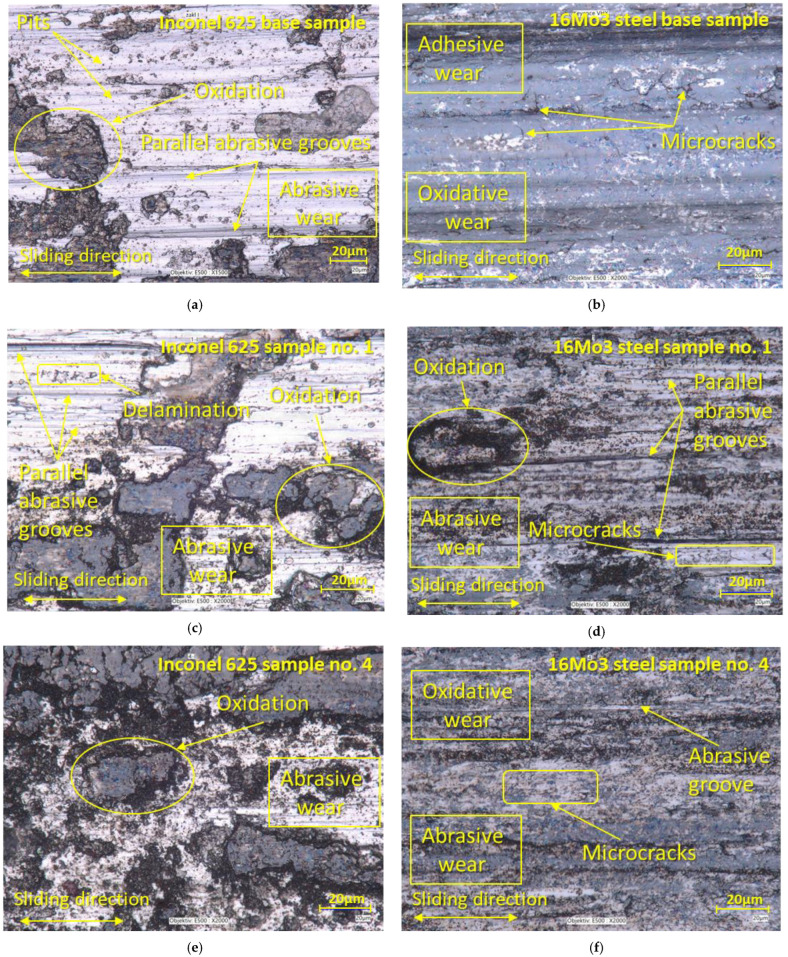
Analysis of surface wear: (**a**) Base Inconel 625; (**b**) Base 16Mo3 steel; (**c**) Inconel 625 sample no. 1; (**d**) 16Mo3 steel sample no. 1; (**e**) Inconel 625 sample no. 4; (**f**) 16Mo3 steel sample no. 4.

**Figure 10 materials-16-05014-f010:**
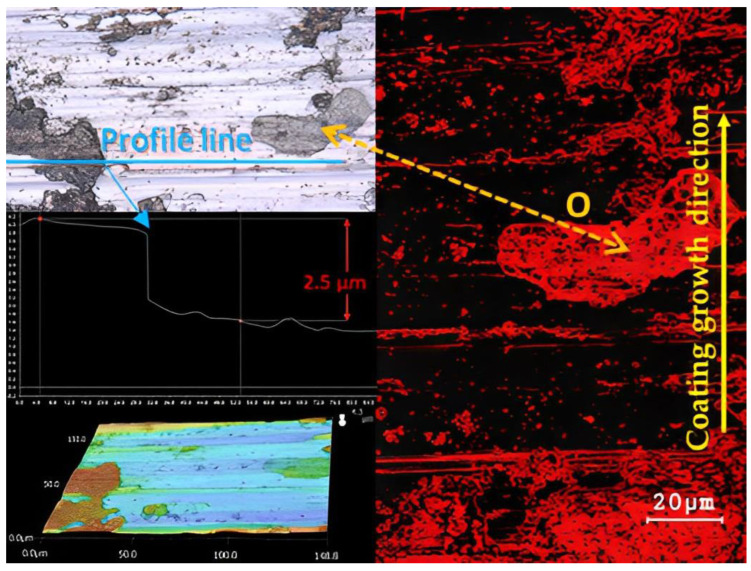
EDS results of Inconel 625 base sample.

**Figure 11 materials-16-05014-f011:**
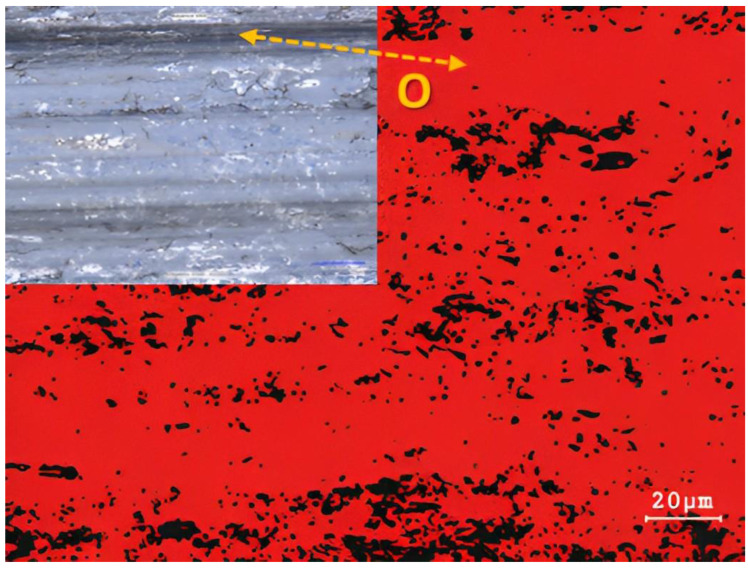
EDS results of 16Mo3 steel base sample.

**Figure 12 materials-16-05014-f012:**
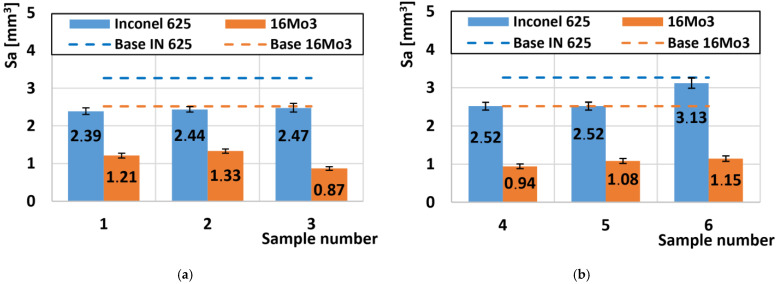
Comparison of roughness Sa: (**a**) samples (no. 1–3) at the outer radius of the pipe; (**b**) samples (no. 4–6) on the inner radius of the pipe.

**Figure 13 materials-16-05014-f013:**
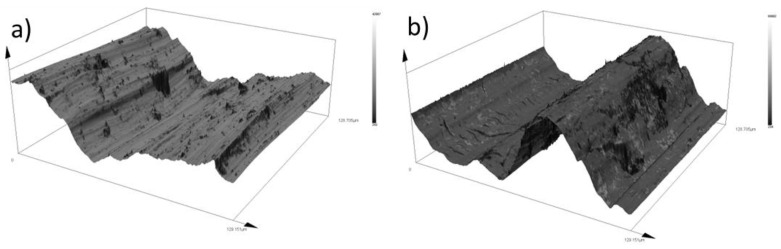
3D comparison of roughness values: (**a**) Inconel 625 base sample, (**b**) 16Mo3 steel base sample, (**c**) Inconel 625 sample no. 1, (**d**) 16Mo3 sample no. 1, (**e**) Inconel 625 sample no. 4 and (**f**) 16Mo3 sample no. 4.

**Table 1 materials-16-05014-t001:** Chemical composition of the Inconel 625 alloy and 16Mo3 steel [wt.%].

Material	Cr	Mo	Co	Nb	Ti	Fe	C	Mn	Si	Al	Ni
**Inconel 625**	24.40	8.57	0.05	4.74	0.20	0.89	0.08	0.01	0.05	0.26	60.30
**16Mo3**	0.02	0.31	-	-	-	98.50	0.19	0.69	0.20	0.02	0.25

## Data Availability

Data are available upon request to the corresponding author.

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
