# Peer review of "Effect of Supercritical Bending on the Mechanical & Tribological Properties of Inconel 625 Welded Using the Cold Metal Transfer Method on a 16Mo3 Steel Pipe"

_materials, 2023, doi:10.3390/ma16145014_

Round 1

Reviewer 1 Report

firstly, according to the presented content in the manuscrit, the title is not
suitable. It is necessary to precise the Inconel 625 has been coated in the
inner and external surface of the steel pipe; the coating process (CMT) should be indicated with complet name bacause it is not a commun process; "Ni-based supperalloy Inconel 625" can be simplified with commercial name "Inconel 625" in the title and precised in abstract;

Secondly, in abstract, it is necessary to precise the coating geometry (in both saide of pipe) and coating thickness; it is aslo very important ot leave some quantitiative description about the main achievement/results;

Thirdly, in the last paragrphe of the introduction part, it is absolutely necessary to indicate explicitely the originality/specificity of the study;

Fourthly, in part 2 (Materials and Methods), the test protocole should be indicated clearly: welding/coating parameters,  sample geometry (for hardness and for tribology test), sample surface preparation, test zone, test number, test precision/error etc...

Fifthly, the results presentaiton should be improved largely to lead a better understanding:

- in caption of figure 1, it is necessary to precise what correspond to the 6 digital numbers;

- in caption  of figure 4, 5, 7, 8, 9, 12 and 13,  it is necessary to precise what correspond to the different analysis zones (refer to figure 1 ???); it is absolutely necessary to add the corresponding test precison/error bar ;

- in caption of figure 2, it is necessary to add the corresponding reference;

- in caption of figure 3, it is improtant to precise the analysis zone refering to the figure 1; in figure 3(a), it is necessary to indicate the coating growth direction in considering to the anisotropic microstructure;

- in figure 6, it is necessary to precise what correspond to the zone A and B;

- in figure 10, it is important to precise the coating growth direction;

Sixthly, before the conlcusion part, it is extremely important to add a sperated discussion part, to leave a global synthesis about the different results and to explain the results evolution;

Reviewer 2 Report

The review article explores the investigation of mechanical tribological properties on Inconel 625 superalloy welded on a 16Mo3 steel pipe through a supercritical bending process, emphasizing the effects of plastic deformation on the materials' properties.

Here are some suggestions for improving the article:

  1. Line 99: Please add a period at the end of the sentence.

  2. Line 108: The mechanical properties are not presented in Tab. 1.

  3. Line 112: The chemical composition requires correction.

  4. Lines 114 and 127: Please include the reference link and the temperature of cross-section in the ternary diagram.

  5. Please provide details about the methods and equipment employed in the study in the "Materials and Methods" section. Transfer Fig. 3a (lines 128-131) to the "Results and Discussion" section.

  6. When citing references, prioritize recent publications within the last five years to encompass the latest research and advancements in the field. Currently, over 76% of the cited references are from before 2019.

General conclusion. The article can be accepted after minor revisions.

Reviewer 3 Report

The paper deals with a mediocre topic. The paper needs an overall improvement on the metallurgical material science level.

Points for the paper:

The wear is properly estimated and the experimental evaluated accordingly.

The application of a high-temperature pipe is in theory a valid application for a bimetallic material.

Points against.

The question is why would the data at room temperature be significant as these materials are clearly intended for high temperatures.

The authors show a very sloppy attitude toward the material characterization, the steel has an incomplete chemical composition for example, Mo is missing, while it is a very important alloying element in the specific steel.

The base microstructure of ferrite and pearlite is so poorly presented, that the two microstructural components are very difficult to distinguish. 

The authors talk about dendritic shape grains, it is certainly a dendritic microstructure, but these cannot be always characterized as single grains. Anyway, the terminology is not at the appropriate level. The interdendritic areas are not clearly characterized. 

The ternary diagram in Figure 2 is a gross oversimplification and should be removed and a proper phase analysis shown. 

Alloying elements are not "alloys".

The austenite grains in 625 are rarely homogeneous and the occurrence of carbides and nitrides is common.

The weld itself is never presented. Does it have a gap? Is it a proper metallurgical bond?

The SEM secondary phase is not properly presented, also is it a secondary phase or did it form during solidification?

HAZ is not presented.

What happened do the seam during deformation? Show micrographs and comment.

The study itself does not show any real relevance. While the authors clearly show knowledge in the field of tribology, they fail to present the relevance of the work.

HV0.05 is a very small load the values of such measurements are much higher than if it was measured by for example HV10, especially in the weld material, as it has a dendritic microstructure.

Some sentences are a bit awkward to read, and some unorthodox expressions are used. The english should be rvised.

Round 2

Reviewer 1 Report

A global discussion between the different results is missing: there are any evident interactins and interconnections between the 4 subsections divided into 4 clear units; each separated results part have any influences between them ? what are the coupling effects between the different results ?

Reviewer 3 Report

The authors have made minor improvements to the paper, but have not addressed all the issues.

1) Material characterization is still not correctly done.

-the micrograph of ferrite and pearlite is better, but could still be improved a bit in terms of image quality (not a major concern),

-the Inconel 625 characterization is poor, if the interdendritic areas are secondary phases, then identify them. The authors need to be aware of the meaning of primary and secondary phase. It is in the nature of how they formed, this is neither cleary presented nor has any proper attempt been made. Stating that they made a literature survey, while not porpery using the data is not enough.

Removal of figure 2 was a suggestion, it can stay, but a proper explanation has to be made. Describe all the phases in the figure caption and a proper presentation in the text, the existing one is not good enough. Make it simple and conciese, while additionally describing the effect of the phases that occur in the diagram.

Please add a section in the introduction describing the heat and temperature resistance of 16Mo and what problems are solved by an Inconel coating. 

No special comment.

Round 3

Reviewer 1 Report

in the revised version of the manuscript, author has made necesssary corrections including all remarks from reviewers.

Reviewer 3 Report

The authors have significantly improved their paper.

The English style needs a bit of polishing, a typical example:"The purpose of  studying these superalloy deposits is to reduce the cost of their creation and extend their  service life." The text is understandable, but a bit odd, the use of "cost of creation" is unusually, production cost for example is better.